# Modelling the demand for new nitrogen fixation by terrestrial ecosystems

Xu-Ri<sup>1,2,\*</sup>, I. Colin Prentice<sup>3</sup>

<sup>1</sup>Key Laboratory of Alpine Ecology and Biodiversity, Institute of Tibetan Plateau Research, Chinese Academy of Sciences, Beijing 100101, China
 <sup>2</sup>CAS Center for Excellence in Tibetan Plateau Earth Sciences, Beijing 100101, China
 <sup>3</sup>AXA Chair of Biosphere and Climate Impacts, Department of Life Sciences, Imperial College London, Silwood Park Campus, Buckhurst Road, Ascot SL5 7PY, UK

Correspondence to: Xu-Ri (xu-ri@itpcas.ac.cn)

Abstract. Continual input of reactive nitrogen (N) is required to support the natural turnover of N in terrestrial ecosystems. This "N demand" can be satisfied in various ways including biological N fixation (BNF) (the dominant pathway under natural conditions), lightning-induced abiotic N fixation, N uptake from sedimentary substrates, and N deposition from natural and anthropogenic sources. We estimated the global new N fixation demand (NNF), i.e. the total new N input

- required to sustain net primary production (NPP) in non-agricultural terrestrial ecosystems regardless of its origin, using a Nenabled global dynamic vegetation model (DyN-LPJ). DyN-LPJ does not explicitly simulate BNF; rather, it estimates total NNF using a mass balance criterion and assumes that this demand is met from one source or another. The model was run in steady state, and then in transient mode driven by recent changes in CO<sub>2</sub> concentration and climate. A range of values for key stoichiometric parameters was considered, based on recently published analyses. Modelled NPP, and C:N ratios of litter
- and soil organic matter, were consistent with independent estimates. Modelled geographic patterns of ecosystem NNF were similar to other analyses, but actual estimated values exceeded recent estimates of global BNF. The results were sensitive to a few key parameters: the fraction of litter carbon respired to  $CO_2$  during decomposition, and plant type-specific C:N ratios of litter and soil. The modelled annual NNF increased by about 15% during the course of the transient run, mainly due to increasing  $CO_2$  concentration. The model did not overestimate recent terrestrial carbon uptake, suggesting that the increase in
- NNF demand has so far been met. Rising  $CO_2$  is further increasing the NNF demand, while the future capacity of N sources to support this is unknown.

## **1** Introduction

Terrestrial plant growth depends on net primary production (NPP), which is what remains of total photosynthetic carbon (C) fixation (gross primary production, GPP) after plant respiration has and other C losses have returned about half of the GPP to the atmospheric carbon dioxide (CO<sub>2</sub>) pool. Global terrestrial NPP is about 50-60 Pg C yr<sup>-1</sup>. NPP is approximately balanced by the transfer of plant matter to detritus (litter), which is decomposed by microbial action to

become soil organic matter (SOM) with the release of much of its C content as  $CO_2$ . Eventually the SOM itself is also oxidized to  $CO_2$ . In steady state, NPP must equal the total release of  $CO_2$  from the decomposition of litter and SOM, plus a small contribution from fire. With rising atmospheric  $CO_2$ , rates of photosynthesis and NPP can increase and therefore C stocks can increase, allowing net uptake of anthropogenic  $CO_2$  (Ciais et al., 2014). However, plant tissues contain elements

- 5 in addition to carbon, hydrogen and oxygen most abundantly nitrogen (N), which originates as N<sub>2</sub> in the atmosphere but must be supplied to plants in reactive forms including nitrate (NO<sub>3</sub><sup>-</sup>) and ammonium (NH<sub>4</sub><sup>+</sup>). N is repeatedly recycled between plants and soil: when inorganic N is released (mineralized) from litter and SOM during decomposition, it becomes available for re-uptake by plants (or microbes). A large fraction of the total N stock in most ecosystems is recycled in this way, The global annual recycled N has been quantified to be ~ 1Pg N (Cleveland et al., 2013;Xu-Ri and Prentice, 2008), .
- But the cycle is not closed. N is lost through leaching (both dissolved and particulate forms are taken along with flows of water in the soil, and transferred to streams and rivers), and as gases: ammonia (NH<sub>3</sub>) emitted by volatilization, and nitric oxide (NO), nitrous oxide (N<sub>2</sub>O) and dinitrogen (N<sub>2</sub>) emitted by microbial processes, principally denitrification. These losses have to be replenished by new supplies of reactive N for a steady NPP to be maintained, and the supply rate has to increase further if NPP and C storage are to increase. We refer to this requirement for new reactive N supplies to terrestrial ecosystems (plant and soil) from any source, not only biological nitrogen fixation (BNF) as the 'new N fixation demand'

(NNF).

There are large uncertainties in current knowledge of the N inputs to terrestrial ecosystems. BNF, and to a lesser extent nitrogen oxide ( $NO_x = NO + NO_2$ ) production from N<sub>2</sub> by lightning, are the main natural processes that can satisfy the N demand of ecosystems. Early estimates of global terrestrial BNF were 90-130 Tg N yr<sup>-1</sup> (Galloway et al., 1995) and 100-290

- Tg N yr<sup>-1</sup> (Cleveland et al., 1999), based on upscaling field measurements. But recent global estimates are much lower, e.g. 58 (40-100) Tg N yr<sup>-1</sup> (Vitousek et al., 2013). (Sullivan et al., 2014) suggested downgrading conventional estimates of BNF in tropical forests (generally regarded as a hotspot of N fixation) by a factor of five, based on new measurements. Early large estimates of the lightning contribution to N fixation (> 100 Tg N yr<sup>-1</sup>: (Liaw et al., 1990) have also been revised downwards, to 1-20 Tg N yr<sup>-1</sup> (Labrador, 2005)., Natural NO<sub>x</sub> emissions from soils (and fires) can be transported in the atmosphere and
- subjected to dry or wet deposition in other places, but this flux to terrestrial ecosystems is small in the preindustrial world: about 4.5 Tg N yr<sup>-1</sup> for oxidized N species (NO<sub>y</sub>) and 13 Tg N yr<sup>-1</sup> for reduced species (NH<sub>x</sub>) (Galloway et al., 1995). Human activities have altered the global N cycle through the widespread use of N fertilizer, whereby atmospheric N<sub>2</sub> is initially fixed by the Haber-Bosch process, and the release of reactive N to the atmosphere through fossil fuel burning. Global agricultural N inputs have been estimated as ~140 Tg N yr<sup>-1</sup> (Galloway et al., 1995;Schlesinger, 2009) and total N deposition
- 30 over land in recent times as ~50 Tg N yr<sup>-1</sup> (Dentener et al., 2006), much larger than the natural N deposition rate. But the fate of most fertilizer N is to be either lost in gaseous emissions, or leached out of the fields and transported away in streams. Enhanced atmospheric N deposition is concentrated near populous industrialized regions, resulting in N saturation or even overload in some places, but with limited effect over most of the global land surface(Cleveland et al., 2013). BNF remains as the largest likely contributor to satisfying terrestrial ecosystems' new N demand in a global perspective, while uncertainty

surrounds the actual magnitudes both of the global new N demand and of the extent to which it is satisfied by BNF. Moreover, rising  $CO_2$  concentration and the resulting increase in GPP have inevitably further increased the new N demand . Thus three key knowledge gaps are (1) the magnitude of the global new N demand; (2) the magnitude of terrestrial BNF, and its ability to satisfy demand; and (3) to what extent, and by what mechanisms, terrestrial ecosystems have been able to respond to  $CO_2$ -induced increases in N demand through the enhanced acquisition of N.

5

- Model-based analyses have not yet cast much light on these issues as there is still no consensus on how to represent the coupling of the terrestrial C and N cycles. The first dynamic global vegetation models (DGVMs) did not consider N cycle processes at all. (Hungate et al., 2003) first drew attention to the large discrepancy between early 'optimistic' DGVM projections of high rates of carbon uptake in a high-CO<sub>2</sub> world (Cramer et al., 2001) and independent projections of N uptake
  based on contemporary rates. This analysis set a value of ~ 90 Tg N yr<sup>-1</sup> for current terrestrial BNF (Galloway et al., 2002). Several recent DGVMs have included strong N supply limitations on both NPP and the response of NPP to increasing CO<sub>2</sub> concentration, yet the process most likely to limit NPP in the long term that is, BNF has been represented in indirect ways: for example, as a function of actual evapotranspiration (Yang et al., 2009;Zaehle and Friend, 2010), based on earlier analyses by (Schimel et al., 1996) and (Cleveland et al., 1999), or simply as a function of NPP (see the discussion by (Wieder et al., 2015)). Some models have prescribed rather than predicted BNF (Houlton et al., 2008;Gerber et al., 2010;Esser et al., 2011). The basis for modelling N inputs to ecosystems thus remains largely unresolved. In this paper, we
- use a mass-balance approach, as implemented in the DyN-LPJ model of Xu-Ri & Prentice (2008), to address the question: how much newly fixed N *must* be made available each year, globally, from any source, in order to sustain NPP ? In other words, what is the 'demand' for newly fixed N for terrestrial ecosystem – and how can it be satisfied, based on current understanding of supply-side constraints?

The DyN-LPJ model of (Xu-Ri & Prentice, 2008), which has also been used to quantify the N<sub>2</sub>O-climate feedback (Xu-Ri *et al.*, 2012; Stocker *et al.*, 2013), takes a different approach from other models. It assumes that annual N fixation by terrestrial ecosystems must not only balance losses of N, but also provide sufficient new N inputs to maintain the observed stoichiometry of plant, litter, decomposer biomass and SOM. The model thus calculates the new N supply based on mass

25 balance considerations that is required to satisfy the N demand of terrestrial ecosystem both from pland and soil. This demand cannot be fully met by recycling (N uptake and immobilization) from the soil inorganic N pool. The calculation involves the C:N ratios of plant litter and SOM and the fraction of litter C that is respired to CO<sub>2</sub>. We make use of recently published analyses of observational and experimental data on these parameters to constrain the demand for fixed N, and we model transient changes in demand based on observed changes in CO<sub>2</sub> concentration and climate.

## 2 Materials and Methods

#### 2.1 Model description

In addition to the coupled carbon and water cycling and vegetation dynamics processes represented in the LPJ dynamic global vegetation model (Sitch *et al.*, 2003)., DyN-LPJ simulates the flows of N through atmosphere, vegetation, litter and soiland back into the atmosphere including submodels for plant N uptake, N allocation, N mineralization from litter and soil, BNF, nitrification, NH<sub>3</sub> volatilization, nitrate leaching, denitrification, and N<sub>2</sub>, N<sub>2</sub>O and NO production and emission (Xu-Ri and Prentice, 2008;Xu-Ri et al., 2012). In the earlier version of DyN-LPJ, however, the inorganic N requirement of microbial growth was met from new input, resulting in an unrealistically high rate of total new N input.

Here we have added a key feature essential for this analysis: namely the representation of immobilization – the uptake of inorganic N into microbial biomass – as a major source of N to fuel decomposition (Fig. 1). The breakdown of complex organic molecules by microbial and mycorrhizal action into soluble, organic forms that can be taken up by plants or microbes – now recognized as an important "bypass" to the soil inorganic N pool(Schimel and Bennett, 2004) – is not represented explicitly, but this should not influence the calculation of NNF.

The full dynamic N mass-balance equations of the model are listed in Appendix S1. All the abbreviations used in the text are described in Table A1. Some insights into the N cycle as represented in DyN-LPJ can be obtained by considering the relationships among modelled N fluxes that would apply in steady state (see Table A1 for symbols and abbreviations). For the total organic N pool (plants, litter and SOM) to be in steady state,

$$NNF + N_{up} + N_{immo} - (f_a \cdot N_{minL} + N_{minS}) = 0$$
<sup>(1)</sup>

where  $N_{up}$  is N uptake by vegetation,  $N_{immo}$  is microbial N uptake (immobilization),  $f_a$  is the 'atmospheric fraction' i.e. the 20 fraction of litter C that is returned to the atmosphere as CO<sub>2</sub> during decomposition, and  $f_a.N_{minL}$  and  $N_{minS}$  are the gross mineralization rates from litter and SOM, respectively. For the soil inorganic N pool to be in steady state,

$$(f_a \cdot N_{minL} + N_{minS}) - N_{up} - N_{immo} - N_{loss} = 0$$

where  $N_{loss}$  is the total loss of N (gaseous losses plus leaching). In steady state NNF =  $N_{loss}$ , so NNF can be found from either equation (1) or equation (2):

25 
$$NNF = N_{loss} = (f_a \cdot N_{minL} + N_{minS}) - N_{up} - N_{immo}$$
(3)

The terms on the right-hand side of equation (3) can now be expressed as follows:

$$f_a \cdot N_{minL} = \operatorname{NPP} f_a / R_{CR} \tag{4}$$

$$N_{minS} = \text{NPP}\left(1 - f_a\right)/R_S \tag{5}$$

$$N_{up} = \text{NPP}/R_P \tag{6}$$

$$N_{immo} = \text{NPP} \left( \frac{1}{R_{CR}} - \frac{1}{R_L} \right) \tag{7}$$

where NPP is net primary production;  $R_S$  is the C:N ratio of SOM; and  $R_P$  is the C:N ratio for plant production, as specified in Table 2. During decomposition, an increase in litter N (net immobilization) may take place before release of litter N (net mineralization) begins. Net mineralization only occurs after litter N concentration has increased to  $R_{CR}$ , the 'critical' C:N

5 ratio, which depends on the C:N ratio of undecomposed litter,  $R_L$  (Parton et al., 2007; Manzoni et al., 2008). The N resorption flux remains within the plant N pool, and therefore does not contribute to NNF. By combining equation (3) with equations (4) to (7) and assuming  $R_P \approx R_L$ , we obtain the following expression for steady-state NNF:

$$NNF \approx NPP \left(1 - f_a\right) \left(1/R_s - 1/R_{CR}\right) \tag{8}$$

showing how NNF depends on the atmospheric fraction and the relative magnitudes of  $R_s$  and  $R_{CR}$ . The composition of 10 undecomposed litter determines  $R_{CR}$  (Parton et al., 2007;Manzoni et al., 2008) according to an empirical formula derived from litter decomposition experiments, given by (Manzoni et al., 2008) as:

$$r_{CR} = 0.45 \ r_L^{0.76} \tag{9}$$

in terms of N:C ratios ( $r_{CR}$  and  $r_L$ ), where  $r_{CR} = 1/R_{CR}$  and  $r_L = 1/R_L$ .

Equation (9) expresses two important functional properties of the decomposer community. First, the kinetics of 15 decomposition are determined by the undecomposed litter chemical composition and do not change as decomposition proceeds. Second, decomposers that can break down carbon-rich litter also have a high critical C:N ratio corresponding to a low carbon use efficiency,  $e = R_B/R_{CR}$  where  $R_B$  is the C:N ratio of the decomposer biomass (Manzoni et al., 2008). Unlike the critical C:N ratio, the microbial biomass C:N ratio relatively conservative along gradients of organic matter or litter C:N, being typically in the range of 5 - 15. The fraction of litter C returned to the atmosphere by respiration is 1 - e.

#### 20 2.2 Climate and CO<sub>2</sub> forcing

A steady-state and a transient model run were set up using identical parameter values, spin-up protocols and forcings to the simulations described by (Xu-Ri et al., 2012) except that the transient run was repeated and extended to 2009, substituting TS 3.10.1 climate data (<u>http://www.cru.uea.ac.uk/cru/data/hrg/</u>) from the Climatic Research Unit, and updated atmospheric CO<sub>2</sub> concentration data from (Keeling et al., 2009), for the input data sets used previously. The contributions of

climate and CO<sub>2</sub> changes to the transient simulation were assessed as in (Xu-Ri et al., 2012) by performing an additional transient run with time-varying climate but constant CO<sub>2</sub> (296 ppm).

### 2.3 Sensitivity and uncertainty analysis

We considered the effect of varying  $R_s$  in the steady-state simulation from 4/5 to 5/4 of our central estimates (Tables 1, 2), a range corresponding to that found in the literature. We also examined the effect of varying *e* in the transient simulation. Many models, including the previously published version of DyN-LPJ, have set *e* = 0.3 (Sitch et al., 2003). This value was

- 5 derived from the DEMETER model (Foley, 1995) and appears to have originated from CENTURY (Parton et al., 1992). Recent experimental determinations have indicated lower values of *e*, for example 0.25 in tropical Amazonian forest (Chambers et al., 2001) and 0.20 in temperate beech forest (Ngao et al., 2005). Assuming  $R_B = 10$ , the default value used by (Manzoni et al., 2008), results in a global average *e* of 0.23. The global average value of  $R_B$  has been estimated as ~7.6 (Xu et al., 2013), so the true global average value of *e* may be even lower (~0.175). Accordingly, we performed alternative model
- 10 runs with  $R_B = 7.6$  (low), 8.6 (intermediate) and 10 (high). The corresponding *e* values are 0.175 (low), 0.2 (intermediate) and 0.23 (high).

### **3 Results**

## 3.1 Steady-state NNF

Global NPP in the steady-state run was 50.8 (49.6-51.3) Pg C yr<sup>-1</sup>, within the generally accepted range (Cramer et al., 15 1999). Total global ecosystem NNF was 340 (230-470) Tg N yr<sup>-1</sup> (Table 1). The geographic distribution of modelled NNF (Fig. 2) shows maxima in tropical forests and savannas, with tropical ecosystems (30 S-30 N) contributing 67% and northern extratropical ecosystems 30% to the global total. Ranges by biome were 4-10 g N m<sup>-2</sup> yr<sup>-1</sup> in tropical ecosystems, 2-4 g N m<sup>-2</sup> yr<sup>-1</sup> in humid subtropical forests, mediterranean-type ecosystems, maritime humid forests and boreal forests, and < 2 g N m<sup>-2</sup> yr<sup>-1</sup> in temperate grasslands, tundra and desert.

- The calculated NNF is influenced by the fraction of litter carbon respired to  $CO_2$  during decomposition and plant functional type (PFT)-specific C:N ratios of litter and soil. Litter C:N ratios in the model are mainly determined by the PFTspecific C:N ratios of production ( $R_P$ , Table 2). The simulated global average litter C:N ratio in the model was 48.9 (Table 1), indistinguishable from 49.9 ± 3 as given in a recent review (Yang and Luo, 2011). The global average estimate of  $R_{CR}$  (~ 43) is close to the value of 40 estimated by Parton *et al.* (2007) and Manzoni *et al.* (2008). The global average modelled soil C:N
- 25 ratio was 15.8 (Table 1), higher than the estimate of 13.3 by Post *et al.* (1985) but close to the recent value of 16.4 (Xu *et al.*, 2013) and lower than the value of 18.5 given by (Yang and Luo, 2011).

Uncertainty analysis of the steady-state run (Tables 1, 2) confirmed our expectation that lower soil C:N ratios ( $R_s$ ) would result in larger modelled NNF while higher values would result in reducedNNF. If our analysis were only based on plant N demand, this might resulted unrealistic high C to N ratio of ~43 for SOM, might not match the realistic values of around 13, 16. This analysis indicated that access the new N input need to maintain the C to N ratio of both plant and soil

The C:N ratios of litter ( $R_L$ ), in contrast, are closely tied to  $R_P$  and vary little among the simulations. A change of  $R_L$  between 48 and 50 (larger than simulated) would only change the critical C:N ratio ( $R_{CR}$ ) from 42 to 43.5 (from eq. 1). Variation in  $R_{CR}$  through a larger range from 40 to 43 (Parton *et al.*, 2007) only results in a change in modelled NNF from 340 to 360 Tg N yr<sup>-1</sup>. This uncertainty range is much smaller than that arising from the uncertainty in  $R_S$ .

## 5 3.2 Changes in NNF in response to changes in CO<sub>2</sub> and climate

Global NPP increased from 42.6 to 52.0 Pg C yr<sup>-1</sup> during the transient simulation. Lower, central and upper estimates of NNF (obtained by setting *e* at 0.175, 0.2 and 0.23) yielded increases through the same period from 290 to 340 Tg N yr<sup>-1</sup>, 340 to 410 Tg N yr<sup>-1</sup>, and 400 to 470 Tg N yr<sup>-1</sup> respectively (Fig. 3a). The increase in NNF was 40 to 60 Tg N yr<sup>-1</sup> (Fig. 3b) depending on the chosen value of *e*. About 80% of this increase was directly caused by the rising CO<sub>2</sub> concentration (Fig. 3a). The rate of increase in modelled NNF amounted to 0.47 to 0.67 Tg N yr<sup>-1</sup> for each ppm increase in CO<sub>2</sub> (Fig. 4d). Altogether about 76% of this additional NNF came from tropical ecosystems and about 17% from the northern extratropics (Fig. 3b), with a spatial pattern highlighting modelled hotspots of "woody thickening" in temperate and tropical savannas and

woodlands (Fig. 5). There was a strong correlation between modelled NNF and NPP, both in terms of spatial ( $R^2 = 0.85$ ) and temporal ( $R^2 = 0.86$ ) patterns (Fig. 4b, c). The slope of the relationship was 0.007 to 0.009 g N g<sup>-1</sup> C.

15

10

#### 3.3 N losses and denitrification

Denitrification accounted for 71% of total modelled N loss. The modelled global denitrification rate, and the total N loss from terrestrial ecosystems, were from 180 to 240 and 260 to 340 TgN yr<sup>-1</sup> respectively (Fig. 3c, d). In the transient simulation, N loss and denitrification rates were positively correlated ( $R^2 = 0.94$ ). Both were more sensitive to climate than to CO<sub>2</sub> concentration (Fig. 3c, d; see also (Xu-Ri et al., 2012). The additional fixed N taken up in response to increasing CO<sub>2</sub>

to  $CO_2$  concentration (Fig. 3c, d; see also (Xu-Ri et al., 2012). The additional fixed N taken up in response to increasing  $CO_2$  concentration was mainly stored in organic forms (Fig. 6a-c): on average 52% in SOM, 30% in litter, and the remainder in plant biomass.

The global terrestrial denitrification rate can be very roughly constrained by global natural land N<sub>2</sub>O emissions, given assumptions about the N<sub>2</sub> to N<sub>2</sub>O ratio in gaseous losses of N. The modeled global N<sub>2</sub>O emission from unfertilized land was previously estimated as 8.6 Tg N yr<sup>-1</sup> (with a range of 7.6 to 10.5 Tg N yr<sup>-1</sup>) (Xu-Ri *et al.*, 2012), constrained by 66 worldwide measurements of total annual N<sub>2</sub>O emissions from natural ecosystems. Modeled N<sub>2</sub> to N<sub>2</sub>O ratios varied between 25 and 50 (Xu-Ri *et al.*, 2012), as determined by the maximum rate of N<sub>2</sub>O production from denitrification in (Xu-Ri & Prentice, 2008). These values fall within the broad range of 20 to 220 from direct measurements of both fluxes made with a

state-of-the-art technique (Dannenmann et al., 2008).

## 3.4 NNF compared to N recycling between plant and soil

The total rate of N recycling from inorganic to organic compartments – equal to N uptake (0.98-1.05 Pg N yr<sup>-1</sup>, Table 1) plus immobilization (0.15 Pg N yr<sup>-1</sup>) – was estimated as 1.13-1.20 Pg N yr<sup>-1</sup>. The reverse flux – equal to mineralization from litter (0.95-0.99 Pg N yr<sup>-1</sup>) and SOM (0.44 to 0.69 Pg N yr<sup>-1</sup>) – was estimated as 1.39-1.68 Pg N yr<sup>-1</sup>. The imbalance between these two fluxes (recycling and mineralization) represents NNF, which has to be met from outside the 'loop' formed by plants and soil (Fig. 1). The modelled steady-state immobilization was 147-151 Tg N yr<sup>-1</sup>, about 10% of the total N mineralization rate (1.39-1.68 Pg N yr<sup>-1</sup>), consistent with experimental results (Hadas et al., 1992).

The modelled NPP to NNF ratio was in the range 110-140 (Fig. 4b,c). This value is much larger than C:N ratio of plant production because much of the N required for plant production is satisfied by recycled N. The fraction of NPP supported by

10 NNF is given by the product of NNF and  $R_P$ /NPP. Globally, the model indicates that NNF supplies only ~30% of the N requirement for plant production, the rest being provided by recycled N – but there is considerable regional variation (Fig.7). It should be noted that the models provides area-average estimates, implicitly including areas where vegetation is recovering from episodic disturbances, which are expected to experience enhanced demand.

## **4** Discussion

5

## 15 4.1 Comparison with previous estimates of BNF

If BNF is assumed to be the largest supplier of N to terrestrial ecosystems, it makes sense to compare our estimated terrestrial ecosystems N demand for new N fixation (NNF) with independent estimates of BNF. However, our central estimate of global terrestrial N demand (340 Tg N yr<sup>-1</sup>) exceeds the *upper bound* of 290 Tg N yr<sup>-1</sup> given by Cleveland *et al.* (1999) for global terrestrial BNF, and exceeds more recent estimates (e.g. 127.5 Tg N yr<sup>-1</sup>, Cleveland *et al.*, 2013; 58 Tg N

- 20 yr<sup>-1</sup>, (Vitousek et al., 2013) by a large factor. Our biome-average model estimates of N demand (Table 1) are similar to upper bounds of BNF given by (Cleveland et al., 1999) (Fig. 4a) while the model estimates generally greater N demand on a siteby-site basis than the (Cleveland et al., 1999) BNF data indicate, especially in high latitudes (Table 3). Thus there is an important gap between our model calculations of the N demand in non-agricultural ecosystems, and most estimates of the supply of newly fixed N through BNF.
- There could be several reasons for this disparity, which we cannot currently distinguish. On the one hand, our model formulation may overestimate the N demand. It would be useful to compare our formulation with alternative modelling approaches to the estimation of total N demand. On the other hand, there is considerable heterogeneity among different estimates of BNF; some agents of BNF may not have been sufficiently considered; and other routes of entry for N may possibly be important. Some recent N fixation measurements based on the <sup>15</sup>N dilution technique have indicated that N
- 30 fixation in alpine and temperate grasslands could be as high as > 1 g N m<sup>-2</sup> yr<sup>-1</sup>, comparable with our estimates of N demand for these ecosystems (Yang et al., 2011). One recent analysis of 99 canopy trees in tropical forest also indicated a high

fixation rate of 8-20 g N m<sup>-2</sup> yr<sup>-1</sup> (Wurzburger and Hedin, 2016), comparable with our estimates of N demand in tropical ecosystems(Table 3). Additional N inputs derived from the weathering of fixed N in sedimentary rocks (Morford et al., 2011) may contribute significantly to meeting ecosystem N demand on deep soils (Mckinley et al., 2009). (Stocker et al., 2016) noted the remarkable diversity of natural N sources and the poor state of quantification of most of them, indicating a need for new field research to attempt to close ecosystem N budgets, especially in tropical ecosystems.

5

## 4.2 The fraction of NPP supported by newly fixed N

(Cleveland et al., 2013) provided estimates of the fraction of terrestrial NPP that is supported by newly fixed N, noting that an anlogous concept of 'new production' is well established in biological oceanography. They used satellite data to derive NPP and a method based on published syntheses of field measurements to derive the fraction of NPP supported by symbiotic and asymbiotic N fixation and N deposition. They estimated a total recycled N flux of 1.05 Pg N yr<sup>-1</sup>, similar to 10 our estimated range of 0.98-1.05 Pg N yr<sup>-1</sup> (Table 1). Our modelled fraction of NPP supported by new fixed N in tropical ecosystems is much higher than in temperate and boreal forests(Fig. 7), in broad agreement with (Cleveland et al., 2013). However we estimated a larger fraction of total global NPP to be dependent on new N inputs (~30%, as opposed to 11% in Cleveland et al., 2013) due to our larger estimate of global ecosystem N demand(NNF).

- Resorption from senescent leaves is an important pathway of nutrient recycling in most terrestrial ecosystems. Because 15 resorbed N remains in the plant N pool and is subequently re-allocated during bud formation and early leaf expansion, increased N availability in soil might result in decreased N resorption (Brant and Chen, 2015;Lu et al., 2013). (Cleveland et al., 2013) estimated that about 30% of plant N demand was met by resorption. However, the N resorption flux remains within the plant N pool, and therefore does not contribute to the satisfaction of NNF as we define it. The impact of assuming that 30% of plant N uptake is obtained from resorption is illustrated by the cyan numbers in Fig. 1, whereby the plant N 20

uptake decreases, initial C:N ratio of litter and N immobilization increases but NNF is unchanged.

#### 4.3 Has rising N demand been met?

The 'residual land sink' – that is, the uptake of  $CO_2$  by those land ecosystems that have not been losing carbon due to deforestation – is estimated to have been  $2.6 \pm 1.2$  Pg C yr<sup>-1</sup> during both the 1990s and the 2000s (Ciais et al., 2014), based on top-down calculations that are independent of terrestrial models. With C:N ratios for terrestrial organic matter in the 25 range of 30 to 70 (De Vries et al., 2008; Sutton et al., 2008) it follows that the terrestrial N store must have increased at about 40 to 90 Tg N yr<sup>-1</sup>. This is consistent with our model estimates of a C:N ratio in the range of 35 to 50 (R<sub>E</sub>,Table 1) and an increased NNF by 40 to 60 Tg N yr<sup>-1</sup>, with the additional N stored mainly in organic pools. Ciais et al. (2014) also drew attention to the need for increased N inputs to match terrestrial carbon uptake while maintaining stoichiometric constraints.

The rates of carbon uptake by the land during the 1990s and 2000s were modelled (central estimates) by DyN-LPJ as 1.7 and 1.8 Pg C yr<sup>-1</sup> respectively. Thus, the model *underestimated* the residual land sink. The rate of increase in the modelled terrestrial demand for N amounted to 0.47 -0.67 Tg N yr<sup>-1</sup> for each ppm increase in CO<sub>2</sub> (Fig. 4d). Presumably, this

increasing demand for N has been met, or exceeded, at a global scale; otherwise the observed terrestrial C uptake could not have occurred. This conclusion admits the possibility of increasing N limitation on NPP in some ecosystems, such as boreal forests, but nonethless poses a question as to the origin of the additional fixed N required to support carbon uptake on land.

## 4.4 N limitation and anthropogenic influences

- It has been hypothesized that BNF might increase by 10-45% with  $CO_2$  doubling (Hungate et al., 2003), but some experiments have suggested that increasing plant growth might not be sustained over many years of  $CO_2$  elevation (Hungate et al., 2004) because of the limitation of BNF and/or plant biomass accumulation by supplies of other elements. Strong N limitation of NPP has been reported in temperate and boreal forests (De Vries et al., 2006) and even in tropical forests (LeBauer and Treseder, 2008), while limited N supply has been mentioned frequently as a constraint on the  $CO_2$  fertilization
- effect and has recently been shown to be a strong constraint on biomass increase in ecosystems dominated by arbiscular mycorrhizal symbioses (Terrer et al., 2016). On the other hand, 'mysterious N sources' have been invoked to sustain the increased carbon uptake by forests under long-term CO<sub>2</sub> enrichment (Mckinley et al., 2009). To some extent, CO<sub>2</sub>-driven increases in NPP as observed in Free Air Carbon dioxide Enrichment (FACE) experiments may have been supported by increased exploration of the soil and increased rates of total N mineralization from SOM (Drake et al., 2011). (Zaehle et al.,
- 2014) noted that the key process by which plants can acquire additional N to support CO<sub>2</sub>-enhanced growth under N-limited conditions, as shown in some FACE experiments, is enhanced 'mining' of N from SOM. They found this to be a neglected process in DGVMs, with some models succeeding in reproducing observed CO<sub>2</sub>-enhanced growth but for the wrong reason, i.e. due to an unrealistic degree of flexibility in the C:N ratio of plant biomass. But SOM 'mining' is presumably a process that has a time limit as potential N supplies in SOM are finite, reflecting the accumulation of a fraction of the N acquired by
- the ecosystem over time.

One non-mysterious source of newly fixed N is anthropogenic N deposition, which may have a synergistic effect with  $CO_2$  in promoting enhanced NPP in temperate forests (Lloyd, 1999). Modelled NNF increased by 13-17% (average 15%) with increasing  $CO_2$  (Fig. 3b), composed of 22-34 Tg N yr<sup>-1</sup> in the tropics and 13-19 Tg N yr<sup>-1</sup> in the northern extratropics. According to (Dentener, 2006), atmospheric N deposition over land during the 1990s amounted to 22.5 Tg N yr<sup>-1</sup> in the tropics and 27.5 Tg N yr<sup>-1</sup> in the northern extratropics. Anthropogenic N deposition is thus of a large enough magnitude to have contributed significantly to satisfying increased NNF. However, its geographic distribution is extremely patchy. Most tropical and many temperate forests are remote from the large anthropogenic sources. When we compare the N supply by atmospheric N deposition (Dentener, 2006) with the modelled increase in NNF (Fig. 5) in the regions of heaviest N deposition (Europe, North America, South and East Asia) it appears that there is already an overload of N, i.e. more N is

30 deposited than can be stored by organic components, in these regions; while other regions remain N-limited (Fig. 8).

## **5** Concluding remarks

Many authors have drawn attention to the need for increased N inputs to match terrestrial carbon uptake while maintaining the stoichiometry of plant and microbial life. Rising  $CO_2$  concentration continues to increase natural ecosystems' demand for N at a global scale. Over multi-millennial time scales, it appears that new N inputs can increase sufficiently to

- support large increases in land carbon storage driven by increasing atmospheric CO<sub>2</sub> concentration, as took place over the last glacial-interglacial transition (Prentice et al., 2011). But the rate at which such adaptation can take place is unknown. Given the discrepancy between our mass-balance calculations and recent estimates of the rate at which newly fixed N enters the land biosphere, and considerable uncertainties surrounding this quantity, our impression is that current understanding of the sources of fixed N is insufficient to allow reliable process-based modelling of these sources. This discrepancy cannot
- plausibly be accounted for entirely by N deposition or mining of N from SOM. The extent to which the supply of newly fixed N can increase in response to increasing N demand is likewise unclear, and this knowledge gap remains an important uncertainty in model projections of the global C cycle. To address it will require consideration of both the assumptions and implications of alternative numerical schemes to predict N demand, and empirical research to better quantify the components of total ecosystem N budgets.

#### Author contributions

X-R developed the model, performed the model simulations and evaluations, and wrote successive drafts. ICP participated in the model development, analysis and writing.

## 20 Acknowledgments

This research was funded by the National Natural Science Foundation of China (40975096, 41175128,41575152), Strategic Priority Research Program—Climate Change: Carbon Budget and Related Issues of the Chinese Academy of Sciences (XDA05020402, XDA05050404-3-2). It is a contribution to the AXA Chair programme on Biosphere and Climate Impacts and the Grand Challenges in Ecosystems and the Environment initiative at Imperial College London.

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

Table 1 Modeled global NNF in steady state, including the range due to uncertainty in the soil C:N ratio (steady-state runs with e = 0.175). NNF, ecosystem demand for newly fixed N; N<sub>immo</sub>, N immobilization rate; N<sub>up</sub>, N uptake rate; N<sub>min</sub>, N mineralization rate; NPP, net primary production; R<sub>P</sub>, C:N ratio of production; R<sub>v</sub>, C:N ratio of vegetation; R<sub>L</sub>, C:N ratio of litter; R<sub>s</sub>, C:N ratio of soil organic matter; R<sub>E</sub>, C:N ratio of ecosystems.

| Experiment                 | NNF<br>(Tg N yr <sup>-</sup><br><sup>1</sup> ) | N <sub>immo</sub><br>(Tg N yr <sup>-</sup><br><sup>1</sup> ) | N <sub>up</sub><br>(Pg N yr <sup>-1</sup> ) | N <sub>min</sub><br>(Pg N yr <sup>-1</sup> ) | NPP<br>(Pg C yr <sup>-1</sup> ) | R <sub>P</sub> | R <sub>v</sub> | R <sub>L</sub> | R <sub>S</sub> | R <sub>E</sub> |
|----------------------------|------------------------------------------------|--------------------------------------------------------------|---------------------------------------------|----------------------------------------------|---------------------------------|----------------|----------------|----------------|----------------|----------------|
| $1 \times central$         | 337.3                                          | 150.2                                                        | 1.025                                       | 1.54                                         | 50.78                           | 49.50          | 187.9          | 48.90          | 15.82          | 42.04          |
| estimate of $R_S$          |                                                |                                                              |                                             |                                              |                                 |                |                |                |                |                |
| $4/5 \times central$       | 471.6                                          | 150.6                                                        | 1.050                                       | 1.68                                         | 51.26                           | 48.80          | 182.4          | 48.50          | 12.99          | 35.35          |
| estimate of $R_S$          |                                                |                                                              |                                             |                                              |                                 |                |                |                |                |                |
| $5/4 \times central$       | 227.6                                          | 147.8                                                        | 0.983                                       | 1.39                                         | 49.63                           | 50.49          | 183.4          | 49.29          | 19.65          | 50.82          |
| estimate of R <sub>S</sub> |                                                |                                                              |                                             |                                              |                                 |                |                |                |                |                |

| PFT                                | R <sub>P</sub> | R <sub>s</sub> | R <sub>s</sub> | R <sub>S</sub> |
|------------------------------------|----------------|----------------|----------------|----------------|
|                                    |                | (central       | (4/5 ×         | (5/4 ×         |
|                                    |                | estimate)      | central        | central        |
|                                    |                |                | estimate)      | estimate)      |
| Tropical Broad-leaved Evergreen    | 43.75          | 16.73          | 13.38          | 20.91          |
| Tropical Broad-leaved Raingreen    | 32.66          | 8.31           | 6.65           | 10.39          |
| Temperate Needle-leaved Evergreen  | 89.17          | 23.86          | 19.09          | 29.83          |
| Temperate Broad-leaved Evergreen   | 90.63          | 25.78          | 20.62          | 32.23          |
| Temperate Broad-leaved Summergreen | 65.00          | 20.09          | 16.07          | 25.11          |
| Boreal Needle-leaved Evergreen     | 52.38          | 29.70          | 23.76          | 37.13          |
| Boreal Needle-leaved Summergreen   | 45.24          | 18.15          | 14.52          | 22.69          |
| Temperate Herbaceous               | 54.29          | 9.77           | 7.82           | 12.21          |
| Tropical Herbaceous                | 69.55          | 10.34          | 8.27           | 12.93          |

Table 2 Prescribed C:N ratios for plant production  $(R_P)$  and soil organic matter  $(R_S)$  (McGuire et al., 1992;Xu-Ri and Prentice, 2008)

| Vegetation types                                                | Longitude      | Latitude    | Location       | Simulated              | Range of N fixation    |  |  |
|-----------------------------------------------------------------|----------------|-------------|----------------|------------------------|------------------------|--|--|
|                                                                 |                |             |                | NNF                    | rates in (Cleveland et |  |  |
|                                                                 |                |             |                | $(g N m^{-2} yr^{-1})$ | al., 1999)             |  |  |
|                                                                 |                |             |                |                        | $(g N m^{-2} yr^{-1})$ |  |  |
| Moist tundra and                                                | alpine tundr   | a           |                |                        |                        |  |  |
|                                                                 | -145.5         | 65.5        | Alaska         | 2.40                   | 0.28 to 0.94           |  |  |
|                                                                 | -113.5         | 53.5        | Canada         | 1.67                   |                        |  |  |
|                                                                 | 16.5           | 62.5        | Sweden         | 1.20                   |                        |  |  |
| Average                                                         |                |             |                | 1.76                   | 0.94                   |  |  |
| Boreal forest and                                               | boreal wood    | land        |                |                        |                        |  |  |
|                                                                 | 19             | 65          | Sweden         | 1.29                   | 0.1 to 0.3             |  |  |
|                                                                 | 11.5           | 64          | Norway         | 0.96                   |                        |  |  |
|                                                                 | 26.5           | 63          | Finland        | 1.13                   |                        |  |  |
| Average                                                         |                |             |                | 1.13                   | 0.196                  |  |  |
| Temperate conife                                                | rous forest, d | leciduous f | orest and mixe | ed forest              |                        |  |  |
|                                                                 | -114           | 50          | Rocky          | 1.94                   | 0.1 to 16              |  |  |
|                                                                 |                |             | Mountains      |                        |                        |  |  |
|                                                                 | -89            | 51          | Ontario,       | 1.30                   |                        |  |  |
|                                                                 |                |             | Canada         |                        |                        |  |  |
|                                                                 | 12             | 47.5        | Austria        | 1.58                   |                        |  |  |
|                                                                 | 175            | -41         | New            | 3.15                   |                        |  |  |
|                                                                 |                |             | Zealand        |                        |                        |  |  |
| Average                                                         |                |             |                | 1.99                   | 2.658                  |  |  |
| Temperate savanna, temperate tall grassland and short grassland |                |             |                |                        |                        |  |  |
|                                                                 | -93            | 45.5        | USA            | 1.42                   | 0.1 to 1               |  |  |
|                                                                 | -96.5          | 37          | Oklahoma,      | 2.86                   |                        |  |  |
|                                                                 |                |             | USA            |                        |                        |  |  |
|                                                                 | -105           | 41          | Colorado,      | 1.38                   |                        |  |  |
|                                                                 |                |             | USA            |                        |                        |  |  |
| Average                                                         |                |             |                | 1.89                   | 0.305                  |  |  |

Table 3 Site-by-site comparison of modeled NNF (steady-state run, 340 ppm CO<sub>2</sub>, with e = 0.175) with biological N fixation data summarized in (Cleveland et al., 1999).

| Tropical savanna and wet savanna |               |             |              |      |              |  |  |
|----------------------------------|---------------|-------------|--------------|------|--------------|--|--|
|                                  | 28.5          | -24.5       | South Africa | 2.66 | 0.07 to 3.45 |  |  |
|                                  | -6.5          | 7.5         | Ivory coast  | 6.53 |              |  |  |
|                                  | 6.5           | 9           | Nigeria      | 4.82 |              |  |  |
| Average                          |               |             |              | 4.67 | 4.400        |  |  |
| Arid shrublands                  |               |             |              |      |              |  |  |
|                                  | -113          | 41          | Utah, USA    | 1.33 | 3 to 9.75    |  |  |
|                                  | -68           | -34         | Argentina    | 1.18 |              |  |  |
|                                  | -100.5        | 30.5        | Southwest    | 3.06 |              |  |  |
|                                  |               |             | USA          |      |              |  |  |
| Average                          |               |             |              | 1.86 | 3.393        |  |  |
| Tropical evergree                | n forest      |             |              |      |              |  |  |
|                                  | 146.5         | -7.5        | New Guinea   | 6.60 | 0.1 to 24.3  |  |  |
|                                  | -72.5         | 3.5         | Colombia     | 6.58 |              |  |  |
|                                  | 80.5          | 8.5         | Sri Lanka    | 6.66 |              |  |  |
|                                  | -156          | 19.5        | Hawaii       | 4.13 |              |  |  |
| Average                          |               |             |              | 5.99 | 3.607        |  |  |
| <b>Tropical nonfores</b>         | ted floodplai | n           |              |      |              |  |  |
|                                  | -53           | -9          | Brazil       | 7.40 | 0.63 to 24.3 |  |  |
| Average                          |               |             |              | 7.40 | 5.38         |  |  |
| Tropical deciduou                | s forest and  | tropical wo | odland       |      |              |  |  |
|                                  | -1            | 6           | Kade, Ghana  | 6.92 | 0.75 to 1.76 |  |  |
|                                  | 83            | 25.5        | Chakia,      | 4.33 |              |  |  |
|                                  |               |             | India        |      |              |  |  |
| Average                          |               |             |              | 5.62 | 3.393        |  |  |
| Desert                           |               |             |              |      |              |  |  |
|                                  | -117.5        | 35          | Mojave       | 2.38 | 1 to 10      |  |  |
|                                  | -111.5        | 29.5        | Sonoran      | 1.55 |              |  |  |
|                                  | -117          | 40          | Great Basin  | 1.93 |              |  |  |
|                                  | 130           | -20.5       | Australia    | 2.16 |              |  |  |
|                                  | 22            | -23         | Kalahari     | 1.90 |              |  |  |
| Average                          |               |             |              | 2.00 | 1.078        |  |  |
|                                  |               |             |              |      |              |  |  |

## **Figure captions**

Figure 1 Schematic of stocks flows of N in steady state, as modeled by DyN-LPJ.

Figure 2 Geographic distribution of the modeled terrestrial ecosystems demand for newly fixed N (NNF, g N m<sup>-2</sup> yr<sup>-1</sup>).

**Figure 3** Transient simulations during the 20<sup>th</sup> century, with e = 0.175 and changes in CO<sub>2</sub> and climate, or climate alone: (a) Demand for newly fixed N (NNF, Tg N yr<sup>-1</sup>) (b) Increase in NNF due to rising CO<sub>2</sub> (by latitude bands) (c) Total N loss (d) Denitrification rate.

**Figure 4** Modelled demand for newly fixed N, with e = 0.175: (a) Comparison of biome-average estimates with *upper* bound values from Cleveland *et al.* (1999) (b) Spatial relationship of NNF with NPP (c) Temporal relationship of NNF with

10 NPP during the  $20^{\text{th}}$  century (d) Relationship of increased in global NNF to atmospheric CO<sub>2</sub> concentration.

Figure 5 Geographic distribution of the increase in NNF due to rising  $CO_2$  (g N m<sup>-2</sup> yr<sup>-1</sup>).

**Figure 6** Transient simulations during the  $20^{th}$  century, with e = 0.175 and changes in CO<sub>2</sub> and climate, or climate alone: (a) Ecosystem N balance (b) Organic N pool (c) Inorganic N pool.

Figure 7 Geographic distribution of the percentage of NPP supported by newly fixed N.

**Figure 8** Excess of atmospheric N deposition over NNF during the 1990s (g N m<sup>-2</sup> yr<sup>-1</sup>). Positive values imply N overload, negative values N limitation. The block structure is due to the coarse resolution of the N deposition input.

#### Appendix S1. Dynamic N balance equations in DyN-LPJ

(1) 
$$dN_{plant}/dt = N_{up} - N_{litterfall}$$
  
(2)  $dN_{litter}/dt = N_{litterfall} + N_{immo} - N_{minL}$   
(3)  $dN_{soil\_organic}/dt = NNF+ (1-f_a) N_{minL} - N_{minS}$ 

(4) 
$$dN_{soil\_inorganic}/dt = f_a N_{minL} + N_{minS} - N_{up} - N_{immo} - N_{los};$$

In steady state:

5

$$N_{minL} = f_a N_{minL} + (1-f_a) N_{minL}$$

N<sub>minL</sub> is the gross mineralization from litter, f<sub>a</sub>N<sub>minL</sub> is the fraction of N in decomposed litter entering the soil inorganic nitrogen pool, and (1-f<sub>a</sub>) N<sub>minL</sub> is the fraction of N in decomposed litter entering the soil organic matter pool. N<sub>minS</sub> is the gross mineralization from soil. NNF, is the ecosystem demand for newly fixed N.

$$\label{eq:solution} \begin{split} dN_{organic\_pool}/dt & = dN_{plant}/dt + dN_{litter}/dt + dN_{soil\_organic}/dt \\ dN_{organic\_pool}/dt & = 0 \end{split}$$

Combining (1) to (3), we obtain:

(5) NNF + 
$$N_{up}$$
 +  $N_{immo}$  -  $f_a N_{minL}$  -  $N_{minS} = 0$ 

(6)
$$N_{minL} = NPP/R_{CR}$$

(7) 
$$N_{minS} = NPP (1 - f_a)/R_S$$
  
(8)  $N_{up} = NPP /R_P$   
(9)  $N_{immo} = NPP (1/R_{CR} - 1/R_L)$   
(10)  $R_P \approx R_L$ 

Combining (5) to (10):

(11) NNF = 
$$(f_a N_{minL} + N_{minS}) - N_{up} - N_{immo}$$
, or

(12) NNF = NPP 
$$(1 - f_a)(1/R_s - 1/R_{CR})$$

For transient conditions Eq. (12) can be written as:

(13) NNF = NPP 
$$(1 - f_a)(1/R_S - 1/R_{CR}) + dN_{\text{organic_pool}}/dt$$

| Table A1 | Definition | of abbre | eviation, | values and | units |
|----------|------------|----------|-----------|------------|-------|
|----------|------------|----------|-----------|------------|-------|

| Abbreviation               | Explanations                                                         | Range of values    | Units                 |
|----------------------------|----------------------------------------------------------------------|--------------------|-----------------------|
| NNF                        | Terrestrial ecosystem new N demand                                   | 230 - 470          | Tg N yr <sup>-1</sup> |
| NPP                        | net primary production                                               | ~50                | Pg C yr <sup>-1</sup> |
| NEE                        | net ecosystem exchange                                               | ~2-3               | Pg C yr <sup>-1</sup> |
| $\mathbf{N}_{\text{immo}}$ | N immobilization rate                                                | ~150               | Tg N yr <sup>-1</sup> |
| $N_{up}$                   | Plant N uptake rate                                                  | ~1.0               | Pg N yr <sup>-1</sup> |
| $N_{minL}$                 | N mineralization rate from litter                                    | ~0.96              | Pg N yr <sup>-1</sup> |
| $N_{minS}$                 | N mineralization rate from SOM                                       | ~0.54              | Pg N yr <sup>-1</sup> |
| N <sub>loss</sub>          | N losses as N gases and leaching                                     | 260 - 340          | Tg N yr <sup>-1</sup> |
| N <sub>litterfall</sub>    | N loss as litter fall                                                | ~1.0               | Pg N yr <sup>-1</sup> |
| N <sub>plant</sub>         | N storage in the plant compartment                                   | ~ 5.3              | Pg N                  |
| N <sub>litter</sub>        | N storage in the litter compartment                                  | ~ 4.6              | Pg N                  |
| $N_{soil\_organic}$        | N storage in SOM                                                     | ~ 56.8             | Pg N                  |
| $N_{soil\_inorganic}$      | N storage in soil inorganic forms                                    | ~ 0.94             | Pg N                  |
| $\mathbf{f}_{\mathbf{a}}$  | the fraction of litter carbon respired to $\ensuremath{\text{CO}}_2$ | 0.825-0.77         |                       |
|                            | during decomposition                                                 |                    |                       |
| e                          | carbon use efficiency of decomposers;                                | 0.175-0.23 in this | 0.3 in (Sitch et      |
|                            | $e = (1 - f_a) = R_B/R_{CR}$                                         | study;             | al., 2003)            |
| R <sub>B</sub>             | C:N ratio of decomposer biomass                                      | 5-15               | (Parton et al.,       |
| R <sub>CR</sub>            | the 'critical' C:N ratio of litter                                   | 40-43              | 2007;Manzoni          |
|                            |                                                                      |                    | et al., 2008)         |
| R <sub>P</sub>             | C:N ratio of production                                              | ~50 (33-91)        | PFT specific          |
| $R_V$                      | C:N ratio of vegetation; $R_V = C_{plant}/N_{plant}$                 | ~180               | Global average        |
| R <sub>L</sub>             | C:N ratio of litter                                                  | ~49                | Table 1               |
| R <sub>S</sub>             | C:N ratio of soil                                                    | 13-19              | Table 1               |
| $R_{\rm E}$                | C:N ratio of ecosystems; $R_E$ =NEE/dNNF                             | 35-51              | Table 1               |