# Peer review of "Modelling the demand for new nitrogen fixation by terrestrial ecosystems"

_Biogeosciences, 2016_

## Referee Comment (RC1) · Anonymous Referee #1 · 16 Nov 2016

This is an interesting and, for the most part, well written paper. It addresses an important and timely question. The analysis is logical and informative. I recommend publication with minor revisions as specified below.

page 2 line2 5-7: It would be good to quantify this recycled N; It is on the order of 98% of the N requirement of NPP in arctic systems, maybe 95% in temperate systems, and I suspect less in tropical systems. On the other hand, disturbance can result in substantial losses that have to be re-accumulated before full recovery. This disturbance-driven loss probably drives most of the NNF. And indeed, outside of the tropics, symbiotic N fixation is usually restricted to early succession.

page 3 lines 11-22; As I interpret this paragraph, the N demand is calculated based on soil N demand alone and does not include plant N demand. I agree that most of

the N is in the soil (except perhaps in tropical systems), but you need to state explicitly that you are ignoring N demand associated with any increase in plant biomass or with changes in plant stoichiometry.

page 4 line 2: I don't think this is true; see Schimel and Bennett 2004 Ecology 85(3):591-602. But the immobilization of inorganic N by microbes is undeniable

page 4 lines 13-14: I am confused by "vary systematically". Surely this ratio changes between bacteria versus fugus dominated decomposition communities

page 4 line 24: what is Rs? Not defined until pater on page 5 line 19

page 5 lines 6-11: these results are interesting in that the NNF seems to vary inversely to the fraction of total N stocks in soil. If the analysis were based on plant N demand, it would suggest that the increased soil ability to meet demand precludes a NNF. However, the analysis is based on soil N demand. Like I said, interesting. Emphasize this point here and expand on it in the discussion.

Throughout; it is difficult for the naive reader to follow all the symbols. A short word description rather than the symbol would make the manuscript easier to read e.g., like you do on page 5 line 19 for Rs (...ooh that's what Rs means) and line 20 for RL

page 6 line 3: these numbers might be easier to interpret if they were expressed as C:N rather N:C ratios. Maybe provide both? These C:N rations are very woody. you might point this out and expand on it in the discussion...Most of the increase in NPP is in woody tissue, as would be expected with a closed canopy?

page 6 lines 6-11; Again I wonder about the role of disturbance in driving N losses from real ecosystems. Even the scattered effects of gap-phase dynamics would add up.

page 6 lines 20-23. I got confused here interpreting "litter" as litter fall or litter production. I'd clarify by changing the wording to "Mineralization from Litter and SOM".

page 6 lines 23-25 again I am confused by the previous description of NNF based on

soil demand and independent of plant demand. Yes I understand that N cycles and anything that goes into the soil will eventually be available to plants. It might just be the wording that has thrown me off. But if I am having trouble interpreting what you did, so will other readers.

page 8 lines 1-8. I think your assessment of resorption versus immobilization in litter works if the vegetation biomass and soil organic mass remain constant, but as fertility increases, I would expect resorption to decline, vegetation biomass to increase and soil organic mass to increase. It is not clear to me that your analysis still holds under those conditions.

Copy editor issues Use of () in citations if the authors name is part of the sentence or not. e.g., page 3 lines5, 32

hanging "this" e.g., page 2 line 22 "but this is a small flux" v. "but this flux is small"

page 3 line 21 this acronym has already been defined on page 2 line 13.

page 3 lines 31-32; the first two sentences of this paragraph are empty and could be deleted by modifying the parenthetical in the next sentence to "described by Xu-Ri and Prentice (2008: see Fig. 1 and Appendix S1)"

page 8 line 26 and perhaps elsewhere: usage "due to" means "caused by" not "because of"

page 9 line 29 "requires" to "require"

---

## Referee Comment (RC2) · Anonymous Referee #2 · 7 Feb 2017

Overall I think this paper is well written, and that the content of the paper is of interest to the readers of biogeosciences. I do have some questions and remarks that I will address below:

In the paper, the authors use a-1 rather than yr-1. This may be confusing?

Introduction:

On page 2, line 24-28, the authors give numbers of N deposition, but the numbers are different while it is unclear what the difference is. E.g. Ndep over land of 50 Tg N a-1 is rather different from the 17,5 gT N a-1 (oxidized and reduced species combined)

Page 3, line 21-22, this sentence is unclear.

Methods:

[Figure]

I think the methods could be worked out further. At times it is unclear what the authors mean, and they go quite quickly through the material. A table with abbreviations, range of values, units, etc would be very helpful.

Also, it was unclear to me what exactly the initial litter chemical composition was, where is the equation that uses this? It is also unclear what rs is, since it is only explained in table 1 and not in the text.

Results:

The results are clear, but the figures are less clear (especially figure 3 and 4). The manuscript would benefit from a better quality graphics.

Page 5, line 27-28 are unclear to me.

Page 7, line 2. How do the authors get to 30%?

Discussion:

I especially like the discussion. The relevant literature is cited, and the paper really comes together at this point.

Some additional small points:

Page 2, line 13. BNF is defined at line 15

Page 3, line 13: dot after ))

Page 4, line 7: take out s from decompositions

Page 4, line 8: input rather than inputs

Page 5, line 1. Where is rs defined?

Page 7, line 14: Table 2 should be Table 3?

---

## Author Response (AR1)

*RC* – Reviewer's Comments, *AC* – Authors' Comments.

*AC*: We greatly thank two Anonymous Referees for providing constructive comments, which are important for improving our manuscript. The comments were carefully evaluated. Based on the comments, we have revised the manuscript. The detailed responses to the comments are shown as below (reviewer's comments in black, authors' responses in blue).

**Responses to Anonymous Referee #1**

RC: This is an interesting and, for the most part, well written paper. It addresses an important and timely question. The analysis is logical and informative. I recommend publication with minor revisions as specified below.

page 2 line2 5-7: It would be good to quantify this recycled N; It is on the order of 98% of the N requirement of NPP in arctic systems, maybe 95% in temperate systems, and I suspect less in tropical systems. On the other hand, disturbance can result in substantial losses that have to be re-accumulated before full recovery. This disturbance-driven loss probably drives most of the NNF. And indeed, outside of the tropics, symbiotic N fixation is usually restricted to early succession.

*AC*: The recycled N has been quantified to be ~ 1 PgN (Cleveland et al., 2013; Xu-Ri and Prentice, 2008), but according to this study, when considering N immobilization, the recycled N might be larger than 1 PgN , ranged 1.13-1.20 Pg N $a^{-1}$. (As shown in Page 6, section 3.4, line 29-31, or Figure 1). Yes, according to figure 7 in this study, In the temperate steppe and boreal forest more than 80% of required N are supported by the recycled N. In the tropical ecosystems, about 70% of the N requirement are supported by the recycled N. While in some extreme cold (arctic), arid and high altitude ecosystems (Tibetan Plateau), more than 50% of the required N were supported by the new N instead of recycled.

Cleveland, C. C., Houlton, B. Z., Smith, W. K., Marklein, A. R., Reed, S. C., Parton, W., Del Grosso, S. J., and Running, S. W.: Patterns of new versus recycled primary production in the terrestrial biosphere, Proceedings of the National Academy of Sciences, 110, 12733-12737, 2013.

Xu-Ri, and Prentice, I. C.: Terrestrial nitrogen cycle simulation with a dynamic global vegetation model, Global Change Biology, 14, 1745-1764, 2008.

RC: page 3 lines 11-22; As I interpret this paragraph, the N demand is calculated based on soil N demand alone and does not include plant N demand. I agree that most of the N is in the soil (except perhaps in tropical systems), but you need to state explicitly that you are ignoring N demand associated with any increase in plant biomass or with changes in plant stoichiometry.

*AC*: As shown the explicit equation in Appendix S1, here we are considering the ecosystem N demand according to the mass balance, both of the plant and soil N demand have been considered. In the revised MS, this paragraph has been revised to be clearer.

RC: page 4 line 2: I don't think this is true; see Schimel and Bennett 2004 Ecology 85(3):591-602. But the immobilization of inorganic N by microbes is undeniable

*AC*: Yes, in this study, we did not considering the soluble organic forms of N that were taken up by plant, for example, monomers. However the ecosystem N demand might not be influenced by either monomer was considered or not. The mass balance figure might be changed as following when the monomers were considered(See attached N fixation chart-R1).

[Figure]

RC: page 4 lines 13-14: I am confused by "vary systematically". Surely this ratio changes between bacteria versus fugus dominated decomposition communities.

*AC*: $R_B$ (C:N ratio of the decomposer biomass) does not like $R_{CR}$ vary systematically along gradients of organic matter and litter N:C and typically remains in the range of 5

to 15, with an assumed average value of 10 in (Manzoni *et al.*, 2008). While in this study, as shown in Page 4 line 10-11, we analyzing the range of the $R_B$ effect on NNF with $R_B$ = 7.6 (low), 8.6 (intermediate) and 10 (high). The corresponding *e* values are 0.175 (low), 0.2 (intermediate) and 0.23 (high). The results can be seen in Page 6, section 3.2, line 5-7, NNF might varing from 290 to 340 Tg N a$^{-1}$, 340 to 410 Tg N a$^{-1}$, and 400 to 470 Tg N a$^{-1}$ respectively (Fig. 3a).

RC: page 4 line 24: what is Rs? Not defined until pater on page 5 line 19

*AC*: Rs is soil C:N ratio, the interpretation has been added when it first appeared in the revised paper.

RC: page 5 lines 6-11: these results are interesting in that the NNF seems to vary inversely to the fraction of total N stocks in soil. If the analysis were based on plant N demand, it would suggest that the increased soil ability to meet demand precludes a NNF. However, the analysis is based on soil N demand. Like I said, interesting. Emphasize this point here and expand on it in the discussion.

*AC*: The NNF here is the ecosystems total N demand. Yes, it is very interesting that we found the NNF are inversely to correlated to the C:N ratio of soil organic matter. The more nitrogen stocks in soil and the more ecosystem NNF demand

"…If the analysis were based on plant N demand, it would suggest that the increased soil ability to meet demand precludes a NNF…", however if the analysis based on plant N demand, this might resulted very high C to N ratio for soil organic matter(might be closer to C to N ratio of litter, ~43), and the resulted high C to N ratio might not match the realistic values of around 13-16. So we listed out in this paper that ecosystem new N input need to maintain the C to N ratio of both plants and soils. .

RC:Throughout; it is difficult for the naive reader to follow all the symbols. A short word description rather than the symbol would make the manuscript easier to read e.g., like you do on page 5 line 19 for Rs (...ooh that's what Rs means) and line 20 for RL.

*AC*: $R_L$ are C to N ratio of litter. In the revised MS, table A1 has been added to describe all the abbreviations.

RC:page 6 line 3: these numbers might be easier to interpret if they were expressed as C:N rather N:C ratios. Maybe provide both? These C:N rations are very woody. you might point this out and expand on it in the discussion...

*AC*: We are using the C:N ratio(R) throughout the paper, only using the N:C ratio(r) with the Manzoni's equation to define the critical C:N ratio.

NPP to NNF ratio are ranged between 110-150, while C to N ratio of plant production were ranged between 33-90 as shown in Table 2. This is mainly because parts of N

required by NPP production are supplied by the recycled N.

RC: page 6 lines 6-11; Again I wonder about the role of disturbance in driving N losses from real ecosystems. Even the scattered effects of gap-phase dynamics would add up.

*AC*: Yes, the disturbed ecosystems, N losses will decrease if without additional fertilizations, for example cropland, might need more new N to feed the losses.

RC: page 6 lines 20-23. I got confused here interpreting "litter" as litter fall or litter production. I'd clarify by changing the wording to "Mineralization from Litter and SOM".

*AC*: litter means the mass pool of the litter component. Around 52% of the new NNF are stored in SOC, 30% in litter and the remaining 28% in living plant.

RC: page 6 lines 23-25 again I am confused by the previous description of NNF based on soil demand and independent of plant demand. Yes I understand that N cycles and anything that goes into the soil will eventually be available to plants. It might just be the wording that has thrown me off. But if I am having trouble interpreting what you did, so will other readers.

*AC*: NNF here is the ecosystem N demand, contains N demand both from the plant and soil

RC: page 8 lines 1-8. I think your assessment of resorption versus immobilization in litter works if the vegetation biomass and soil organic mass remain constant, but as fertility increases, I would expect resorption to decline, vegetation biomass to increase and soil organic mass to increase. It is not clear to me that your analysis still holds under those conditions.

*AC*: I think there are two phases :(1) before NNF demand is satisfied and (2) after NNF were satisfied. During the first phase, with the increasing fertility, the vegetation and soil organic biomass increasing with fertility increase; while on the second phase, with the increasing fertility, vegetation and soil organic biomass will not increase anymore, and soil inorganic N might accumulated with the increasing $NO_3^-$, ecosystem N saturated.

RC: Copy editor issues Use of () in citations if the authors name is part of the sentence or not. e.g., page 3 lines5, 32 hanging "this" e.g., page 2 line 22 "but this is a small flux" v. "but this flux is small"
page 3 line 21 this acronym has already been defined on page 2 line 13.
page 3 lines 31-32; the first two sentences of this paragraph are empty and could be deleted by modifying the parenthetical in the next sentence to "described by Xu-Ri and Prentice (2008: see Fig. 1 and Appendix S1)"

page 8 line 26 and perhaps elsewhere: usage "due to" means "caused by" not "because of"

page 9 line 29 "requires" to "require"

*AC*: All revised.

**Responses to Anonymous Referee #2**

RC: Overall I think this paper is well written, and that the content of the paper is of interest to the readers of biogeosciences. I do have some questions and remarks that I will address below:

In the paper, the authors use a-1 rather than yr-1. This may be confusing?

*AC*: Yes, in the revised paper the a-1 has been changed to yr-1 thoroughly.

RC: On page 2, line 24-28, the authors give numbers of N deposition, but the numbers are different while it is unclear what the difference is. E.g. Ndep over land of 50 Tg N a-1 is rather different from the 17,5 gT N a-1 (oxidized and reduced species combined).

*AC*: The17,5 gT N a-1 is the preindustrial numbers, while the 50 Tg N a-1 is the number of 1990s. In the revised paper these has been clarified.

RC: Page 3, line 21-22, this sentence is unclear.

*AC*: Yes, this sentence has been revised to be clear.

RC: I think the methods could be worked out further. At times it is unclear what the authors mean, and they go quite quickly through the material. A table with abbreviations, range of values, units, etc would be very helpful. Also, it was unclear to me what exactly the initial litter chemical composition was, where is the equation that uses this? It is also unclear what rs is, since it is only explained in table 1 and not in the text.

*AC*: In the revised MS, we enriched the method part, and a table (Table A1) about all the abbreviations was added. The initial litter chemical composition is the initial C to N ratio of litter fall, The C and N in the litter fall are the C and N that were transferred from leaves and roots to litter in the model according the specific turnover rate as shown in (Xu-Ri and Prentice, 2008). Rs is the C to N ratio of soil organic matter. It has been added in its first arise in the text.

RC: Page 5, line 27-28 are unclear to me.

*AC*: NNF are ultimately determined by the following equation (as shown in Appendix S1):

$$NNF = NPP\,(1 - f_a)(1/R_S - 1/R_{CR})$$

the lower soil C:N ratios ($R_S$) would resulted modeled larger NNF, while higher Rs might resulted less NNF. This sentence has been rewrote in the revised MS.

RC: Page 7, line 2. How do the authors get to 30%?

*AC*: The fraction of NPP supported by new fixed N, was given by the product of NNF with the ratio $R_P$/NPP, accounts for a global averge of ~30% of current NPP.

RC: I especially like the discussion. The relevant literature is cited, and the paper really comes together at this point. Some additional small points:

Page 2, line 13. BNF is defined at line 15
Page 3, line 13: dot after ))
Page 4, line 7: take out s from decompositions
Page 4, line 8: input rather than inputs
Page 5, line 1. Where is rs defined?
Page 7, line 14: Table 2 should be Table 3?

*AC*: All revised.

[revised manuscript text omitted]

$$N_{up} = NPP/R_P \tag{6}$$

$$N_{immo} = NPP\, (1/R_{CR} - 1/R_L) \tag{7}$$

where NPP is net primary production; $R_S$ is the C:N ratio of SOM; and $R_P$ is the C:N ratio for plant production, as specified in Table 2. During decomposition, an increase in litter N (net immobilization) may take place before release of litter N (net mineralization) begins. Net mineralization only occurs after litter N concentration has increased to $R_{CR}$, the 'critical' C:N ratio, which depends on the C:N ratio of undecomposed litter, $R_L$ (Parton et al., 2007; Manzoni et al., 2008). The N resorption flux remains within the plant N pool, and therefore does not contribute to NNF. By combining equation (3) with equations (4) to (7) and assuming $R_P \approx R_L$, we obtain the following expression for steady-state NNF:

$$NNF \approx NPP\, (1 - f_a)(1/R_S - 1/R_{CR}) \tag{8}$$

showing how NNF depends on the atmospheric fraction and the relative magnitudes of $R_S$ and $R_{CR}$. The composition of undecomposed litter determines $R_{CR}$ (Parton et al., 2007; Manzoni et al., 2008) according to an empirical formula derived from litter decomposition experiments, given by (Manzoni et al., 2008) as:

$$r_{CR} = 0.45\, r_L^{0.76} \tag{9}$$

in terms of N:C ratios ($r_{CR}$ and $r_L$), where $r_{CR} = 1/R_{CR}$ and $r_L = 1/R_L$.

~~During decomposition, an increase in litter N (net immobilization) may take place before release of litter N (net mineralization) begins. Typically, net mineralization only occurs after litter N concentration increased to a critical value. The *initial* chemical composition of the litter determines the critical C:N ratio ($R_{CR}$) at which this shift takes place (
[revised manuscript text omitted]
50 gas feedbacks from the land biosphere under future climate change scenarios, Nature Clim. Change, 3, 666-672, 10.1038/nclimate1864

http://www.nature.com/nclimate/journal/v3/n7/abs/nclimate1864.html#supplementary-information, 2013.

Stocker, B. D., Prentice, I. C., Cornell, S. E., Davies-Barnard, T., Finzi, A. C., Franklin, O., Janssens, I., Larmola, T., Manzoni, S., Nasholm, T., Raven, J. A., Rebel, K. T., Reed, S., Vicca, S., Wiltshire, A., and Zaehle, S.: Terrestrial nitrogen cycling in Earth system models revisited, New Phytologist, 210, 1165-1168, 10.1111/nph.13997, 2016.

Sullivan, B. W., Smith, W. K., Townsend, A. R., Nasto, M. K., Reed, S. C., Chazdon, R. L., and Cleveland, C. C.: Spatially robust estimates of biological nitrogen (N) fixation imply substantial human alteration of the tropical N cycle, Proceedings of the National Academy of Sciences, 111, 8101-8106, 10.1073/pnas.1320646111, 2014.

Sutton, M. A., Simpson, D., Levy, P. E., Smith, R. I., Reis, S., Van Oijen, M., and de Vries, W. I. M.: Uncertainties in the relationship between atmospheric nitrogen deposition and forest carbon sequestration, Global Change Biology, 14, 2057-2063, 2008.

Terrer, C., Vicca, S., Hungate, B. A., Phillips, R. P., and Prentice, I. C.: Mycorrhizal association as a primary control of the CO2 fertilization effect, Science, 353, 72-74, 10.1126/science.aaf4610, 2016.

Vitousek, P. M., Menge, D. N., Reed, S. C., and Cleveland, C. C.: Biological nitrogen fixation: rates, patterns and ecological controls in terrestrial ecosystems, Philosophical Transactions of the Royal Society B: Biological Sciences, 368, 2013.

Wieder, W. R., Cleveland, C. C., Lawrence, D. M., and Bonan, G. B.: Effects of model structural uncertainty on carbon cycle projections: biological nitrogen fixation as a case study, Environmental Research Letters, 10, 044016

10.1088/1748-9326/10/4/044016, 2015.

Wurzburger, N., and Hedin, L. O.: Taxonomic identity determines N-2 fixation by canopy trees across lowland tropical forests, Ecology letters, 19, 62-70, 10.1111/ele.12543, 2016.

Xu-Ri, and Prentice, I. C.: Terrestrial nitrogen cycle simulation with a dynamic global vegetation model, Global Change Biology, 14, 1745-1764, 2008.

Xu-Ri, Prentice, I. C., Spahni, R., and Niu, H. S.: Modelling terrestrial nitrous oxide emissions and implications for climate feedback, New Phytologist, 196, 472-488, 2012.

Xu, X., Thornton, P. E., and Post, W. M.: A global analysis of soil microbial biomass carbon, nitrogen and phosphorus in terrestrial ecosystems, Global Ecology and Biogeography, 22, 737-749, 2013.

Yang, B., Qiao, N., Xu, X., and Ouyang, H.: Symbiotic nitrogen fixation by legumes in two Chinese grasslands estimated with the 15N dilution technique, Nutrient Cycling in Agroecosystems, 91, 91-98, 2011.

Yang, X., Wittig, V., Jain, A. K., and Post, W.: Integration of nitrogen cycle dynamics into the Integrated Science Assessment Model for the study of terrestrial ecosystem responses to global change, Global Biogeochemical Cycles, 23, GB4029, doi:4010.1029/2009GB003474, 2009.

Yang, Y., and Luo, Y.: Carbon: nitrogen stoichiometry in forest ecosystems during stand development, Global Ecology and Biogeography, 20, 354-361, 2011.

Zaehle, S., and Friend, A. D.: Carbon and nitrogen cycle dynamics in the O-CN land surface model: 1. Model description, site-scale evaluation, and sensitivity to parameter estimates, Global Biogeochemical Cycles, 24, 2010.

Zaehle, S., Medlyn, B. E., De Kauwe, M. G., Walker, A. P., Dietze, M. C., Hickler, T., Luo, Y. Q., Wang, Y. P., El-Masri, B., Thornton, P., Jain, A., Wang, S. S., Warlind, D., Weng, E. S., Parton, W., Iversen, C. M., Gallet-Budynek, A., McCarthy, H., Finzi, A. C., Hanson, P. J., Prentice, I. C., Oren, R., and Norby, R. J.: Evaluation of 11 terrestrial carbon-nitrogen cycle models against observations from two temperate Free-Air CO2 Enrichment studies, New Phytologist, 202, 803-822, 10.1111/nph.12697, 2014.

**Table 1 Modeled global NNF in steady state, including the range due to uncertainty in the soil C:N ratio (steady-state runs with $e = 0.175$). NNF, ecosystem demand for newly fixed N; $N_{immo}$, N immobilization rate; $N_{up}$, N uptake rate; $N_{min}$, N mineralization rate; NPP, net primary production; $R_P$, C:N ratio of production; $R_v$, C:N ratio of vegetation; $R_L$, C:N ratio of litter; $R_S$, C:N ratio of soil organic matter; $R_E$, C:N ratio of ecosystems.**

| Experiment | NNF (Tg N yr$^{-1}$) | $N_{immo}$ (Tg N yr$^{-1}$) | $N_{up}$ (Pg N yr$^{-1}$) | $N_{min}$ (Pg N yr$^{-1}$) | NPP (Pg C yr$^{-1}$) | $R_P$ | $R_v$ | $R_L$ | $R_S$ | $R_E$ |
|---|---|---|---|---|---|---|---|---|---|---|
| 1 × central estimate of $R_S$ | 337.3 | 150.2 | 1.025 | 1.54 | 50.78 | 49.50 | 187.9 | 48.90 | 15.82 | 42.04 |
| 4/5 × central estimate of $R_S$ | 471.6 | 150.6 | 1.050 | 1.68 | 51.26 | 48.80 | 182.4 | 48.50 | 12.99 | 35.35 |
| 5/4 × central estimate of $R_S$ | 227.6 | 147.8 | 0.983 | 1.39 | 49.63 | 50.49 | 183.4 | 49.29 | 19.65 | 50.82 |

**Table 2 Prescribed C:N ratios for plant production ($R_P$) and soil organic matter ($R_S$) (McGuire et al., 1992;Xu-Ri and Prentice, 2008)**

| PFT | $R_P$ | $R_S$ (central estimate) | $R_S$ (4/5 × central estimate) | $R_S$ (5/4 × central estimate) |
|---|---|---|---|---|
| Tropical Broad-leaved Evergreen | 43.75 | 16.73 | 13.38 | 20.91 |
| Tropical Broad-leaved Raingreen | 32.66 | 8.31 | 6.65 | 10.39 |
| Temperate Needle-leaved Evergreen | 89.17 | 23.86 | 19.09 | 29.83 |
| Temperate Broad-leaved Evergreen | 90.63 | 25.78 | 20.62 | 32.23 |
| Temperate Broad-leaved Summergreen | 65.00 | 20.09 | 16.07 | 25.11 |
| Boreal Needle-leaved Evergreen | 52.38 | 29.70 | 23.76 | 37.13 |
| Boreal Needle-leaved Summergreen | 45.24 | 18.15 | 14.52 | 22.69 |
| Temperate Herbaceous | 54.29 | 9.77 | 7.82 | 12.21 |
| Tropical Herbaceous | 69.55 | 10.34 | 8.27 | 12.93 |

**Table 3 Site-by-site comparison of modeled NNF (steady-state run, 340 ppm $CO_2$, with $e$ = 0.175) with biological N fixation data summarized in (Cleveland et al., 1999).**

| Vegetation types | Longitude | Latitude | Location | Simulated NNF ($g\ N\ m^{-2}\ yr^{-1}$) | Range of N fixation rates in (Cleveland et al., 1999) ($g\ N\ m^{-2}\ yr^{-1}$) |
|---|---|---|---|---|---|
| **Moist tundra and alpine tundra** | | | | | |
| | −145.5 | 65.5 | Alaska | 2.40 | 0.28 to 0.94 |
| | −113.5 | 53.5 | Canada | 1.67 | |
| | 16.5 | 62.5 | Sweden | 1.20 | |
| Average | | | | 1.76 | 0.94 |
| **Boreal forest and boreal woodland** | | | | | |
| | 19 | 65 | Sweden | 1.29 | 0.1 to 0.3 |
| | 11.5 | 64 | Norway | 0.96 | |
| | 26.5 | 63 | Finland | 1.13 | |
| Average | | | | 1.13 | 0.196 |
| **Temperate coniferous forest, deciduous forest and mixed forest** | | | | | |
| | −114 | 50 | Rocky Mountains | 1.94 | 0.1 to 16 |
| | −89 | 51 | Ontario, Canada | 1.30 | |
| | 12 | 47.5 | Austria | 1.58 | |
| | 175 | −41 | New Zealand | 3.15 | |
| Average | | | | 1.99 | 2.658 |
| **Temperate savanna, temperate tall grassland and short grassland** | | | | | |
| | −93 | 45.5 | USA | 1.42 | 0.1 to 1 |
| | −96.5 | 37 | Oklahoma, USA | 2.86 | |
| | −105 | 41 | Colorado, USA | 1.38 | |
| Average | | | | 1.89 | 0.305 |

**Tropical savanna and wet savanna**

|  |  |  |  |  |
|---|---|---|---|---|
| 28.5 | −24.5 | South Africa | 2.66 | 0.07 to 3.45 |
| −6.5 | 7.5 | Ivory coast | 6.53 |  |
| 6.5 | 9 | Nigeria | 4.82 |  |
| Average |  |  | 4.67 | 4.400 |

**Arid shrublands**

|  |  |  |  |  |
|---|---|---|---|---|
| -113 | 41 | Utah, USA | 1.33 | 3 to 9.75 |
| -68 | -34 | Argentina | 1.18 |  |
| -100.5 | 30.5 | Southwest USA | 3.06 |  |
| Average |  |  | 1.86 | 3.393 |

**Tropical evergreen forest**

|  |  |  |  |  |
|---|---|---|---|---|
| 146.5 | −7.5 | New Guinea | 6.60 | 0.1 to 24.3 |
| −72.5 | 3.5 | Colombia | 6.58 |  |
| 80.5 | 8.5 | Sri Lanka | 6.66 |  |
| −156 | 19.5 | Hawaii | 4.13 |  |
| Average |  |  | 5.99 | 3.607 |

**Tropical nonforested floodplain**

|  |  |  |  |  |
|---|---|---|---|---|
| −53 | −9 | Brazil | 7.40 | 0.63 to 24.3 |
| Average |  |  | 7.40 | 5.38 |

**Tropical deciduous forest and tropical woodland**

|  |  |  |  |  |
|---|---|---|---|---|
| −1 | 6 | Kade, Ghana | 6.92 | 0.75 to 1.76 |
| 83 | 25.5 | Chakia, India | 4.33 |  |
| Average |  |  | 5.62 | 3.393 |

**Desert**

|  |  |  |  |  |
|---|---|---|---|---|
| −117.5 | 35 | Mojave | 2.38 | 1 to 10 |
| −111.5 | 29.5 | Sonoran | 1.55 |  |
| −117 | 40 | Great Basin | 1.93 |  |
| 130 | −20.5 | Australia | 2.16 |  |
| 22 | −23 | Kalahari | 1.90 |  |
| Average |  |  | 2.00 | 1.078 |

**Figure captions**

**Figure 1** Schematic of stocks flows of N in steady state, as modeled by DyN-LPJ.

**Figure 2** Geographic distribution of the modeled terrestrial ecosystems demand for newly fixed N (NNF, g N m$^{-2}$ yr$^{-1}$).

**Figure 3** Transient simulations during the 20$^{th}$ century, with $e = 0.175$ and changes in $CO_2$ and climate, or climate alone: **(a)** Demand for newly fixed N (NNF, Tg N yr$^{-1}$) **(b)** Increase in NNF due to rising $CO_2$ (by latitude bands) **(c)** Total N loss **(d)** Denitrification rate.

**Figure 4** Modelled demand for newly fixed N, with $e = 0.175$: **(a)** Comparison of biome-average estimates with *upper bound* values from Cleveland *et al*. (1999) **(b)** Spatial relationship of NNF with NPP **(c)** Temporal relationship of NNF with NPP during the 20$^{th}$ century **(d)** Relationship of increased in global NNF to atmospheric $CO_2$ concentration.

**Figure 5** Geographic distribution of the increase in NNF due to rising $CO_2$ (g N m$^{-2}$ yr$^{-1}$) **.**

**Figure 6** Transient simulations during the 20$^{th}$ century, with $e = 0.175$ and changes in $CO_2$ and climate, or climate alone: **(a)** Ecosystem N balance **(b)** Organic N pool **(c)** Inorganic N pool.

**Figure 7** Geographic distribution of the percentage of NPP supported by newly fixed N.

**Figure 8** Excess of atmospheric N deposition over NNF during the 1990s (g N m$^{-2}$ yr$^{-1}$). Positive values imply N overload, negative values N limitation. The block structure is due to the coarse resolution of the N deposition input.

**Appendix S1. Dynamic N balance equations in DyN-LPJ**

(1) $dN_{plant}/dt = N_{up} - N_{litterfall}$

(2) $dN_{litter}/dt = N_{litterfall} + N_{immo} - N_{minL}$

(3) $dN_{soil\_organic}/dt = $   $+ (1-f_a) N_{minL} - N_{minS}$

(4) $dN_{soil\_inorganic}/dt = f_a N_{minL} + N_{minS} - N_{up} - N_{immo} - N_{los}$;

In steady state:

$N_{minL} = f_a N_{minL} + (1-f_a) N_{minL}$

$N_{minL}$ is the gross mineralization from litter, $f_a N_{minL}$ is the fraction of N in decomposed litter entering the soil inorganic nitrogen pool, and $(1-f_a) N_{minL}$ is the fraction of N in decomposed litter entering the soil organic matter pool. $N_{minS}$ is the gross mineralization from soil. NNF, is the ecosystem demand for newly fixed N.

$dN_{organic\_pool}/dt = dN_{plant}/dt + dN_{litter}/dt + dN_{soil\_organic}/dt$

$dN_{organic\_pool}/dt = 0$

Combining (1) to (3), we obtain:

(5) $NNF + N_{up} + N_{immo} - f_a N_{minL} - N_{minS} = 0$

(6) $N_{minL} = NPP / R_{CR}$

(7) $N_{minS} = NPP (1 - f_a)/R_S$

(8) $N_{up} = NPP / R_P$

(9) $N_{immo} = NPP (1/R_{CR} - 1/R_L)$

(10) $R_P \approx R_L$

Combining (5) to (10):

(11) $NNF = (f_a N_{minL} + N_{minS}) - N_{up} - N_{immo}$ , or

(12) $NNF = NPP (1 - f_a)(1/R_S - 1/R_{CR})$

For transient conditions Eq. (12) can be written as:

(13) $NNF = NPP (1 - f_a)(1/R_S - 1/R_{CR}) + dN_{organic\_pool}/dt$

Table A1 Definition of abbreviation, values and units

| Abbreviation | Explanations | Range of values | Units |
|---|---|---|---|
| NNF | Terrestrial ecosystem new N demand | 230 - 470 | Tg N yr$^{-1}$ |
| NPP | net primary production | ~50 | Pg C yr$^{-1}$ |
| NEE | net ecosystem exchange | ~2-3 | Pg C yr$^{-1}$ |
| $N_{immo}$ | N immobilization rate | ~150 | Tg N yr$^{-1}$ |
| $N_{up}$ | Plant N uptake rate | ~1.0 | Pg N yr$^{-1}$ |
| $N_{minL}$ | N mineralization rate from litter | ~0.96 | Pg N yr$^{-1}$ |
| $N_{minS}$ | N mineralization rate from SOM | ~0.54 | Pg N yr$^{-1}$ |
| $N_{loss}$ | N losses as N gases and leaching | 260 - 340 | Tg N yr$^{-1}$ |
| $N_{litterfall}$ | N loss as litter fall | ~1.0 | Pg N yr$^{-1}$ |
| $N_{plant}$ | N storage in the plant compartment | ~ 5.3 | Pg N |
| $N_{litter}$ | N storage in the litter compartment | ~ 4.6 | Pg N |
| $N_{soil\_organic}$ | N storage in SOM | ~ 56.8 | Pg N |
| $N_{soil\_inorganic}$ | N storage in soil inorganic forms | ~ 0.94 | Pg N |
| $f_a$ | the fraction of litter carbon respired to $CO_2$ during decomposition | 0.825-0.77 | |
| $e$ | carbon use efficiency of decomposers; $e = (1 - f_a) = R_B/R_{CR}$ | 0.175-0.23 in this study; | 0.3 in (Sitch *et al.*, 2003) |
| $R_B$ | C:N ratio of decomposer biomass | 5-15 | (Parton et al., |
| $R_{CR}$ | the 'critical' C:N ratio of litter | 40-43 | 2007;Manzoni et al., 2008) |
| $R_P$ | C:N ratio of production | ~50 (33-91) | PFT specific |
| $R_V$ | C:N ratio of vegetation; $R_V = C_{plant}/N_{plant}$ | ~180 | Global average |
| $R_L$ | C:N ratio of litter | ~49 | Table 1 |
| $R_S$ | C:N ratio of soil | 13-19 | Table 1 |
| $R_E$ | C:N ratio of ecosystems; $R_E$=NEE/dNNF | 35-51 | Table 1 |

Figure 1

[Figure]

Figure 2

[Figure]

Figure 3

[Figure]

Figure 4

[Figure]

Figure 5

[Figure]

Figure 6

[Figure]

Figure 7

[Figure]

Figure 8

[Figure]